# Multi-Scale Representation Learning on Proteins

**Vignesh Ram Somnath**[*]
Dept. of Computer Science
ETH Zurich
vsomnath@ethz.ch

**Charlotte Bunne**[*]
Dept. of Computer Science
ETH Zurich
bunnec@ethz.ch

**Andreas Krause**
Dept. of Computer Science
ETH Zurich
krausea@ethz.ch

## Abstract

Proteins are fundamental biological entities mediating key roles in cellular function and disease. This paper introduces a multi-scale graph construction of a protein – HOLOPROT – connecting surface to structure and sequence. The surface captures coarser details of the protein, while sequence as primary component and structure – comprising secondary and tertiary components – capture finer details. Our graph encoder then learns a multi-scale representation by allowing each level to integrate the encoding from level(s) below with the graph at that level. We test the learned representation on different tasks, (i.) ligand binding affinity (*regression*), and (ii.) protein function prediction (*classification*). On the regression task, contrary to previous methods, our model performs consistently and reliably across different dataset splits, outperforming all baselines on most splits. On the classification task, it achieves a performance close to the top-performing model while using 10x fewer parameters. To improve the memory efficiency of our construction, we segment the multiplex protein surface manifold into *molecular* superpixels and substitute the surface with these superpixels at little to no performance loss.

## 1 Introduction

Protein design and engineering has become a crucial component of pharmaceutical research and development, finding application in a wide variety of diagnostic and industrial settings. Besides understanding the design principles determining structure and function of proteins, current efforts seek to further enhance or discover proteins with properties useful for technological or therapeutic applications. To efficiently guide the search in the vast design space of functional proteins, we need to be able to robustly predict properties of a candidate protein [Yang et al., 2019]. Moreover, understanding role and function of proteins is crucial to study causes and mechanism of human disease [Fessenden, 2017].

To achieve this, representations incorporating the complex nature of proteins are required. Proteins consist of amino acids, organic molecules linked by peptide bonds forming a linear *sequence*. Each of the twenty amino acids carries a unique side chain, giving rise to an incomprehensibly large combinatorial space of possible protein sequences. The primary sequence drives the folding of polymers – a spontaneous process guided by hydrophobic interactions, formation of intramolecular hydrogen bonds, and van der Waals forces into a unique three-dimensional *structure*. The resulting shape and *surface* manifold with rich physiochemical properties carry essential information for understanding function and potential molecular interactions.

Previous methods typically only consider an individual subset within these scales, focusing on either sequence [Öztürk et al., 2018, Hou et al., 2018], three-dimensional structure [Hermosilla et al., 2021, Derevyanko et al., 2018] or surface [Gainza et al., 2020]. Two proteins with similar sequences can fold into entirely different conformations. While these proteins might catalyze the same type of

---

[*]Equal contribution.

35th Conference on Neural Information Processing Systems (NeurIPS 2021).

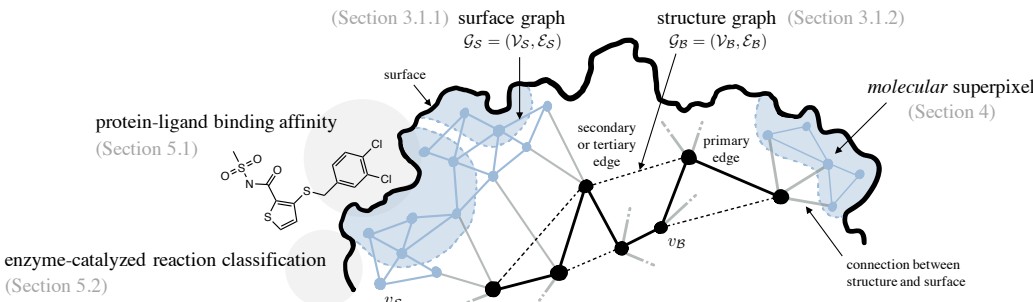

Figure 1: **Overview of HOLOPROT** Our multi-scale protein representation algorithm integrates primary, secondary and tertiary elements of protein structures and connects them to the surface. We extract higher-level protein motifs by introducing *molecular* superpixels. Both structure and surface are represented as graphs $\mathcal{G}_\mathcal{B}$ and $\mathcal{G}_\mathcal{S}$, respectively. The method is evaluated on two representative tasks, protein-ligand binding affinity and enzyme-catalyzed reaction classification.

reactions, their behavior to specific inhibiting drugs might be divergent. Interaction between proteins and ligands, on the other hand, is controlled by molecular surface contacts [Gainza et al., 2020]. Molecular surfaces, determined by subjacent amino acids, are fingerprinted with patterns of geometric and chemical properties, and thus their integration in protein representations is crucial.

In this work, we present a novel multi-scale graph representation which integrates and connects the complex nature of proteins across all levels of information. HOLOPROT consists of a surface and structure layer (both represented as graphs) with explicit edges between the layers. Our construction is guided by the intuition that propagating information from surface to structure would allow each residue to learn encodings reflective of not just its immediate residue neighborhood, but also the higher-level geometric and chemical properties that arise from interactions between a residue and its neighborhood. The associated multi-scale encoder then learns representations by integrating the encoding from the layer below, with the graph at that layer (Section 3). Such multi-scale representations have been previously used in molecular graph generation [Jin et al., 2020] with impressive results.

We further improve the memory efficiency of our construction by segmenting the large and rich protein surface into *molecular "superpixels"*, summarizing higher-level fingerprint features and motifs of proteins. Substituting the surface layer with these superpixels results in little to no performance degradation across the evaluated tasks. The concept of *molecular superpixels* might be of interest beyond our model (Section 4).

The multi-objective and multi-task nature of protein engineering poses a challenge for current methods, often designed and evaluated only on specific subtasks of protein design. By incorporating the biology of proteins, strong representations exhibit robust performance across tasks. We demonstrate our model's versatility and range of applications by deploying it to tasks of rather distinct nature, including a regression task, e.g., inference of protein ligand binding affinity, and classification tasks, i.e., enzyme-catalyzed reaction classification (Section 5).

## 2 Related Work

**Protein Representation Learning**    With increasing availability of sequence and structure data, the field of protein representation learning has advanced rapidly, with methods falling largely in one of the following categories:

*Sequence-based methods.* One-dimensional amino acid sequences continue to be the simplest, most abundant source of protein data and various methods have been developed that borrow architectures developed in natural language processing (NLP). One-dimensional convolutional neural networks have been used to classify a protein sequence into folds and enzyme function [Hou et al., 2018, Dalkiran et al., 2018], and to predict their binding affinity to ligands [Öztürk et al., 2018]. Furthermore, methods have applied complex NLP models trained unsupervised on millions of unlabeled protein sequences and fine-tuned them on different downstream tasks [Rao

et al., 2019, Elnaggar et al., 2020, Bepler and Berger, 2019]. Despite being advantageous when only the sequence is available, these methods ignore the full spatial complexity of proteins.

*Structure-based methods.* To learn beyond sequences, approaches have been developed, that consider the 3D structure of proteins. 3D convolutional neural networks have been utilized for protein quality assessment [Derevyanko et al., 2018], protein contact prediction [Townshend et al., 2019] and protein-ligand binding affinity tasks [Ragoza et al., 2017, Jiménez et al., 2018, Townshend et al., 2020]. An alternate representation treats proteins as graphs, applying graph neural networks for enzyme classification [Dobson and Doig, 2005], interface prediction [Fout et al., 2017], and protein structure quality prediction [Baldassarre et al., 2021]. Gligorijevic et al. [2021] use a long short term memory cell (LSTM) to encode the sequence, followed by a graph convolutional network (GCN) [Kipf and Welling, 2017] to capture the tertiary structure, and apply this to the function prediction task. Hermosilla et al. [2021] propose a convolutional operator that learns to adapt filters based on the primary, secondary, and tertiary structure of a protein, showing strong performance on reaction and fold class prediction.

*Surface-based methods.* Taking a different viewpoint, Gainza et al. [2020] hypothesize that the protein surface displays patterns of chemical and geometric features that fingerprint a protein's interaction with other biomolecules. They utilize geodesic convolutions, which are extensions of convolutions on surfaces, and learn fingerprint vectors, showing improved performance across binding pocket and protein interface prediction tasks.

**Protein Motif Detection** Protein motifs have largely been synonymous with common and conserved patterns in a protein's sequence or structure influencing protein function, e.g., the helix-turn-helix motif binds DNA. Understanding these fragments is essential for 3D structure prediction, modeling, and drug design. While reliably detecting evolutionary motifs, existing tools [Golovin and Henrick, 2008] do not provide a full segmentation of the protein surface manifold. Our work takes a different viewpoint, by looking at protein motifs from the context of a protein surface. Previous methods developed in this context either only consider geometric information rather than physiological properties [Cantoni et al., 2010], are computationally expensive [Cantoni et al., 2011], or designed for particular downstream tasks [Stepniewska-Dziubinska et al., 2020]. Our molecular superpixel approach provides a task-independent segmentation utilizing both geometric and chemical features, while also being computationally efficient.

## 3   Multi-Scale Protein Representation

In this section, we describe our multi-scale graph construction and the associated encoder. Figure 1 illustrates the main principles of HOLOPROT. We represent a protein $\mathcal{P}$ as a graph $\mathcal{G}_{\mathcal{P}}$ with two layers capturing different scales:

(i.) **Surface layer.** This layer captures the coarser representation details of a protein. The protein surface is generated using the triangulation software MSMS [Connolly, 1983, Sanner et al., 1996]. We represent this layer as a graph $\mathcal{G}_{\mathcal{S}}$, where each surface node $u_{\mathcal{S}}$ has a feature vector $\mathbf{f}_{u_{\mathcal{S}}}$ denoting its charge, hydrophobicity and local curvature [Gainza et al., 2020]. Two surface nodes $(u_{\mathcal{S}}, v_{\mathcal{S}})$ have an edge if they are part of a triangulation. Each surface node additionally has a residue identifier $r$, indicating the amino acid residue it corresponds to. Multiple surface nodes can have the same residue identifier.

(ii.) **Structure layer.** This layer captures the finer representation details of a protein. A protein typically has four structural levels: (i.) primary structure (sequence), (ii.) secondary structure ($\alpha$-helices and $\beta$-sheets), (iii.) tertiary structure (3D structure) and (iv.) quaternary structure (complexes) [Fout et al., 2017]. We represent this layer as a graph $\mathcal{G}_{\mathcal{B}}$, where each node $u_{\mathcal{B}}$ corresponds to a residue $r$. Two nodes $(u_{\mathcal{B}}, v_{\mathcal{B}})$ have an edge in $\mathcal{G}_{\mathcal{B}}$ if the $C_{\alpha}$ atoms of the two nodes occur within a certain distance of each other. Distance based thresholding ensures that different structural levels are implicitly captured in the neighborhood of a node $u_{\mathcal{B}}$.

We further introduce edges from the surface layer to the structure layer in order to propagate information between them. Specifically, we introduce a directed edge between a surface node $u_{\mathcal{S}}$ and a backbone node $u_{\mathcal{B}}$ if they both have the same residue identifier $r$. Typically, we have between 20-40 surface nodes $\{u_{\mathcal{S}}\}$ that map to the same structure node $u_{\mathcal{B}}$. This gives us the multi-scale

graph which is then encoded by our multi-scale message passing network. Details on the features used for both the structure and surface layer can be found in Appendix **??**.

## 3.1 Multi-Scale Encoder

Our multi-scale message passing network uses one *message passing neural network (MPN)* for each layer in the multi-scale graph [Lei et al., 2017, Gilmer et al., 2017]. This allows us to learn structured representations of each scale, which can then be tied together through connections between the scales. Before detailing the remainder of the architecture, we introduce some notational preliminaries. For simplicity, we denote the MPN encoding process as $\text{MPN}_\theta(\cdot)$ with parameters $\theta$. We denote $\text{MLP}_\theta(\mathbf{x}, \mathbf{y})$ for a multi-layer perceptron (MLP) with parameters $\theta$, whose input is the concatenation of $\mathbf{x}$ and $\mathbf{y}$, and $\text{MLP}_\theta(\mathbf{x})$ when the input is only $\mathbf{x}$. We also denote the residue identifier of a node $u$ with $\text{id}(u)$, and the neighbors of a node $u$ as $\mathcal{N}(u)$. The details of the MPN architecture are listed in the Appendix **??**.

### 3.1.1 Surface Message Passing Network

We first encode the surface layer $\mathcal{G}_\mathcal{S}$ of the multi-scale protein graph $\mathcal{G}_\mathcal{P}$. The inputs to the MPN are node features $\mathbf{f}_{u_\mathcal{S}}$ and edge features $\mathbf{f}_{u_\mathcal{S} v_\mathcal{S}}$ of $\mathcal{G}_\mathcal{S}$. For more details on the input features used for surface nodes and edges, refer to Appendix **??**. The MPN (with parameters $\theta_\mathcal{S}$) propagates messages between the nodes for $K$ iterations, and outputs a representation $h_{u_\mathcal{S}}$ for each surface node $u_\mathcal{S}$,

$$\{\mathbf{h}_{u_\mathcal{S}}\} = \text{MPN}_{\theta_\mathcal{S}}(\mathcal{G}_\mathcal{S}, \{\mathbf{f}_{u_\mathcal{S}}\}, \{\mathbf{f}_{u_\mathcal{S} v_\mathcal{S}}\}_{v_\mathcal{S} \in \mathcal{N}(u_\mathcal{S})}).$$

### 3.1.2 Structure Message Passing Network

For each node $u_\mathcal{B}$ in the structure layer $\mathcal{G}_\mathcal{B}$, we first prepare the input to the MPN (with parameters $\theta_\mathcal{B}$) by using an MLP (with parameters $\theta$) on the concatenated version of its initial features $\mathbf{f}_{u_\mathcal{B}}$ and the mean of the surface node vectors with the same residue identifier $S = \{\mathbf{h}_{u_\mathcal{S}} | \text{id}(u_\mathcal{S}) = \text{id}(u_\mathcal{B})\}$

$$\mathbf{x}_{u_\mathcal{B}} = \text{MLP}_\theta(\mathbf{f}_{u_\mathcal{B}}, \Sigma_S \, \mathbf{h}_{u_\mathcal{S}}/|S|).$$

Given the edge features $\mathbf{f}_{u_\mathcal{B} v_\mathcal{B}}$, we then run $K$ iterations of message passing, to compute the representations $\mathbf{h}_{u_\mathcal{B}}$ for each structure node $u_\mathcal{B}$,

$$\{\mathbf{h}_{u_\mathcal{B}}\} = \text{MPN}_{\theta_\mathcal{B}}(\mathcal{G}_\mathcal{B}, \{\mathbf{x}_{u_\mathcal{B}}\}, \{\mathbf{f}_{u_\mathcal{B} v_\mathcal{B}}\}_{v_\mathcal{B} \in \mathcal{N}(u_\mathcal{B})}).$$

The graph representation $\mathbf{c}_{\mathcal{G}_\mathcal{P}}$ is an aggregation of structure node representations,

$$\mathbf{c}_{\mathcal{G}_\mathcal{P}} = \sum_{u_\mathcal{B} \in \mathcal{G}_\mathcal{B}} \mathbf{h}_{u_\mathcal{B}}. \tag{1}$$

## 3.2 Task Specific Training

This multi-scale encoding allows us to learn a structured representation of a protein tying different scales together, which can then be utilized for any downstream task. In this work, we evaluate our method on two rather distinct tasks (i.) protein-ligand binding affinity regression, and (ii.) enzyme–catalyzed reaction classification. The architectural details for both downstream tasks are described below. These modules can be adapted and modified in order to utilize HOLOPROT for other use cases.

### 3.2.1 Protein-Ligand Binding Affinity

Protein-ligand binding affinity prediction depends on the interaction of a protein, encoded using the HOLOPROT framework, and a corresponding ligand, in most cases small molecules. To encode the ligand represented as a graph $\mathcal{G}_\mathcal{L}$, we use another MPN (with parameters $\theta_\mathcal{L}$) and aggregate its node representations to obtain a graph representation $c_{\mathcal{G}_\mathcal{L}}$. We concatenate the graph representations $\mathbf{c}_{\mathcal{G}_\mathcal{P}}$ (Equation 1) of the protein and $c_{\mathcal{G}_\mathcal{L}}$ of the ligand, and use that as input to a MLP (with parameters $\phi$) to obtain predictions,

$$s_a = \text{MLP}_\phi(c_{\mathcal{G}_\mathcal{P}}, c_{\mathcal{G}_\mathcal{L}}). \tag{2}$$

The model is trained by minimizing the mean squared error.

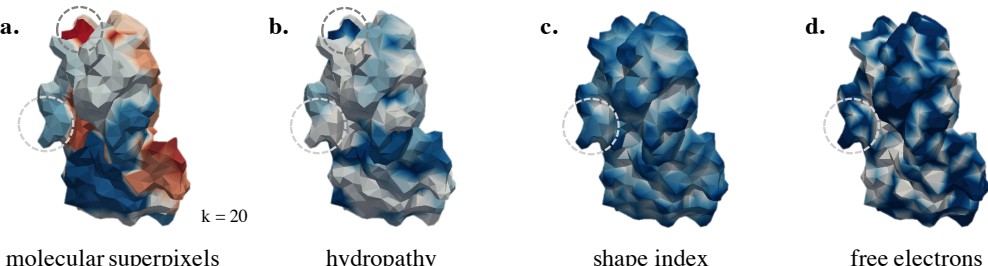

a.

k = 20

molecular superpixels     hydropathy     shape index     free electrons

Figure 2: **Molecular Superpixels and Surface Features of the HIV-1 Protease** (PDB ID: 2AVQ). **a.** Molecular superpixels, indicated by different colors ($k = 20$), and the corresponding surface features, i.e., **b.** hydropathy, **c.** shape index, and **d.** free electrons. As highlighted, molecular superpixels are spatially compact and overlap with surface regions dominated by single features such as hydrophobic patches while capturing coherent areas across all surface features. The protein complex contains 198 residues.

### 3.2.2 Enzyme-Catalyzed Reaction Classification

To predict the enzyme-catalyzed reaction class, we use the graph representation $\mathbf{c}_{\mathcal{G}_{\mathcal{P}}}$ of the protein obtained via HOLOPROT as the input to a MLP (with parameters $\phi$) to obtain the prediction logits,

$$p_k = \mathrm{MLP}_\phi(c_{\mathcal{G}}). \tag{3}$$

The model is trained by minimizing the cross-entropy loss.

## 4 Superpixels on Molecular Surfaces

Protein surface manifolds are complex and represented via large meshes. In order to improve the computational and memory efficiency of our construction, we introduce the notion of *molecular superpixels*. Originally developed in computer vision [Ren and Malik, 2003, Mori et al., 2004, Kohli et al., 2009], superpixels are defined as perceptually uniform regions in the image. In the molecular context, we refer to superpixels as segments on the protein surface capturing higher-level fingerprint features and protein motifs such as hydrophobic binding sites.

In order to apply the segmentation principle to three-dimensional molecular surfaces, we employ graph-based superpixel algorithms on triangulated surface meshes. The superpixel representation of the protein surface needs to satisfy several requirements, as (i.) molecular superpixels should not reduce the overall achievable performance of HOLOPROT, and (ii.) molecular superpixels need to form geometrically compact clusters, and overlap with surface regions that are coherent in physiological surface properties, e.g., capture hydrophobic binding sides or highly charged areas. Popular graph-based segmentation tools such as Felzenszwalb and Huttenlocher [2004, FH], mean shift [Comaniciu and Meer, 2002], and watershed [Vincent and Soille, 1991], however, produce non-compact superpixels of irregular sizes and shapes. By posing the segmentation task as a maximization problem on a graph maximizing over (i.) the entropy rate of the random walk on the surface graph $\mathcal{G}_{\mathcal{S}} = (\mathcal{V}_{\mathcal{S}}, \mathcal{E}_{\mathcal{S}})$ favoring the formation of compact and homogeneous clusters, and (ii.) a balancing term encouraging clusters with similar sizes, the entropy rate superpixel (ERS) segmentation algorithm [Liu et al., 2011] outperforms previous methods across different tasks [Stutz et al., 2018] and achieves the desired properties of *molecular* superpixels.

In order to incorporate geometric and chemical features of the surface $\mathbf{F}_{\mathcal{S}}$, we extend the surface graph $\mathcal{G}_{\mathcal{S}} = (\mathcal{V}_{\mathcal{S}}, \mathcal{E}_{\mathcal{S}})$ with a non-negative similarity measure $w$, given as $w_{ij} = \sum_{\mathbf{f} \in \mathbf{F}_{\mathcal{S}}} |\mathbf{f}_{v_i} \mathbf{f}_{v_j}|$ for nodes $v_i$ and $v_j$ if connected by an edge $e_{ij}$. We simulate a random walk $\mathbf{X} = \{X_t | t \in T, X_t \in \mathcal{V}_{\mathcal{S}}\}$ on a protein surface mesh, where the transition probability $p_{ij}$ between two nodes $v_i$ and $v_j$ is defined as $p_{ij} = P(X_{t+1} = v_j | X_t = v_i) = w_{ij}/w_i$, where $w_i = \sum_{k:e_{ik} \in \mathcal{E}_{\mathcal{S}}} w_{ik}$. The corresponding stationary distributions of nodes $\mathcal{V}_{\mathcal{S}}$ are given by

$$\boldsymbol{\mu} = \left(\mu_1, \mu_2, \ldots, \mu_{|\mathcal{V}_{\mathcal{S}}|}\right)^\top = \left(\frac{w_1}{w_T}, \frac{w_2}{w_T}, \ldots, \frac{w_{|\mathcal{V}_{\mathcal{S}}|}}{w_T}\right)^\top .$$

Molecular superpixels are then defined by a subset of edges $\mathcal{M} \subseteq \mathcal{E}_\mathcal{S}$ such that the resulting graph, $\mathcal{G}_\mathcal{S} = (\mathcal{V}_\mathcal{S}, \mathcal{M})$, contains exactly $k$ connected subgraphs. Computing *molecular* superpixels is achieved via optimizing the objective function with respect to the edge set $\mathcal{M}$

$$\max_\mathcal{M} \underbrace{- \sum_i \mu_i \sum_j p_{ij}(\mathcal{M}) \log\left(p_{ij}(\mathcal{M})\right)}_{\text{(i.) entropy rate}} \underbrace{- \sum_i p_{Z_\mathcal{M}}(i) \log\left(p_{Z_\mathcal{M}}(i)\right) - n_\mathcal{M}}_{\text{(ii.) balancing function}}$$

s.t. $\mathcal{M} \subseteq \mathcal{E}_\mathcal{S}$ and $n_\mathcal{M} \geq k$,

where $n_\mathcal{M}$ is the number of connected components in the graph, $p_{Z_\mathcal{M}}$ denotes the distribution of cluster memberships $Z_\mathcal{M}$, and $\lambda \geq 0$ is the weight of the balancing term. Both terms satisfy monotonicity and submodularity and can thus be efficiently optimized based on techniques from submodular optimization [Nemhauser et al., 1978]. For further details on the entropy rate superpixel algorithm, see Liu et al. [2011].

A molecular superpixel $m$ comprising $k$ surface vertices is then given as $\mathbf{f}_m = (\mathbf{f}_{v_1}, \ldots, \mathbf{f}_{v_k})$ for all $\mathbf{f} \in \mathbf{F}_\mathcal{S}$. We summarize the feature representation of each molecular superpixel via the graph $\mathcal{G}_\mathcal{M} = (\mathcal{V}_\mathcal{M}, \mathcal{E}_\mathcal{M})$, where each node $m \in \mathcal{V}_\mathcal{M}$ is represented via $(\texttt{mean}(\mathbf{f}_m), \texttt{std}(\mathbf{f}_m), \texttt{max}(\mathbf{f}_m), \texttt{min}(\mathbf{f}_m))$ for all $\mathbf{f} \in \mathbf{F}_\mathcal{S}$ and an edge $e \in \mathcal{E}_\mathcal{M}$ via the Wasserstein distance between neighboring superpixels.

Figure 2 demonstrates *molecular* superpixels for the enzyme HIV-1 protease [Brik and Wong, 2003]. Besides being spatially compact, superpixels overlap with surface regions dominated by single features such as hydrophobic patches, while capturing coherent areas across all surface features. Further examples of superpixels are displayed in Appendix **??**.

## 5 Evaluation

Successful protein engineering requires optimization of multiple objectives. When searching for a protein with desired functionality, auxiliary but crucial properties such as stability measured in terms of free energy of folding also need to be satisfied. Furthermore, the field is also subject to a plethora of potential tasks and applications. In order to capture the multi-objective and multi-task nature of protein engineering, we evaluate our method on two representative tasks: regression of the binding affinity between proteins and their ligands, and classification of enzyme proteins based on the type of reaction they catalyze.

### 5.1 Protein-Ligand Binding Affinity Prediction

Studying the interaction between proteins and small molecules is crucial for many downstream tasks, e.g., accelerating virtual screening for potential candidates in drug discovery or protein design to improve the output of an enzyme-catalyzed reaction. The architecture of the regression module is described in Equation 2.

**Dataset.** The PDBBIND database (version 2019) [Liu et al., 2017] is a collection of the experimentally measured binding affinity data for all types of biomolecular complexes deposited in the Protein Data Bank [Berman et al., 2000]. After quality filtering for resolution and surface construction, the refined subset comprises a total of $4,709$ biomolecular complexes. The binding affinity provided in PDBBIND is experimentally determined and expressed in molar units of the inhibition constant ($K_i$) or dissociation constant ($K_d$). Similar to previous methods [Öztürk et al., 2018, Townshend et al., 2020], we do not distinguish both constants and predict negative log-transformed binding affinity $pK_d/pK_i$. We split the dataset into training, test and validation splits based on the scaffolds of the corresponding ligands (*scaffold*), or a 30% and a 60% sequence identity threshold (*identity 30%*, *identity 60%*) to limit homologous ligands or proteins appearing in both train and test sets.

**Baselines.** For evaluating the overall performance on the regression task, we compare HOLOPROT against several baselines including current state-of-the-art methods on both tasks. This comprises sequence-based methods [Öztürk et al., 2018, Rao et al., 2019, Bepler and Berger, 2019, Elnaggar et al., 2020] as well as methods based on the three-dimensional structure of proteins [Townshend et al., 2020, Hermosilla et al., 2021], and recent methods using geometric deep learning on protein molecular surfaces [Gainza et al., 2020].

Table 1: **Protein-Ligand Binding Affinity Prediction Results** Comparison predictive performance of ligand binding affinity using the PDBbind dataset [Liu et al., 2017] of HOLOPROT against other methods. Results are reported for 3 experimental runs.

| Model | # Params | Sequence Identity (30 %) | | | Sequence Identity (60 %) | | |
|---|---|---|---|---|---|---|---|
| | | RMSE | Pearson | Spearman | RMSE | Pearson | Spearman |
| **Sequence-based Methods** | | | | | | | |
| Öztürk et al. [2018] | 1.93 M | $1.866 \pm 0.080$ | $0.472 \pm 0.022$ | $0.471 \pm 0.024$ | $1.762 \pm 0.261$ | $0.666 \pm 0.012$ | $0.663 \pm 0.015$ |
| Bepler and Berger [2019] | 48.8 M | $1.985 \pm 0.006$ | $0.165 \pm 0.006$ | $0.152 \pm 0.024$ | $1.891 \pm 0.004$ | $0.249 \pm 0.006$ | $0.275 \pm 0.008$ |
| Rao et al. [2019] | 93.0 M | $1.890 \pm 0.035$ | $0.338 \pm 0.044$ | $0.286 \pm 0.124$ | $1.633 \pm 0.016$ | $0.568 \pm 0.033$ | $0.571 \pm 0.021$ |
| Elnaggar et al. [2020] | 2.4M[1] | $1.544 \pm 0.015$ | $0.438 \pm 0.053$ | $0.434 \pm 0.058$ | $1.641 \pm 0.016$ | $0.595 \pm 0.014$ | $0.588 \pm 0.009$ |
| **Surface-based Methods** | | | | | | | |
| Gainza et al. [2020] | 0.62 M | $1.484 \pm 0.018$ | $0.467 \pm 0.020$ | $0.455 \pm 0.014$ | $1.426 \pm 0.017$ | $0.709 \pm 0.008$ | $0.701 \pm 0.011$ |
| **Structure-based Methods** | | | | | | | |
| Townshend et al. [2020][2] | - | $\mathbf{1.429 \pm 0.042}$ | $0.541 \pm 0.029$ | $0.532 \pm 0.033$ | $1.450 \pm 0.024$ | $0.716 \pm 0.008$ | $0.714 \pm 0.009$ |
| Townshend et al. [2020][3] | - | $1.936 \pm 0.120$ | $\mathbf{0.581 \pm 0.039}$ | $\mathbf{0.647 \pm 0.071}$ | $1.493 \pm 0.010$ | $0.669 \pm 0.013$ | $0.691 \pm 0.010$ |
| Hermosilla et al. [2021] | 5.80 M | $1.554 \pm 0.016$ | $0.414 \pm 0.053$ | $0.428 \pm 0.032$ | $1.473 \pm 0.024$ | $0.667 \pm 0.011$ | $0.675 \pm 0.019$ |
| HOLOPROT (●) | 1.44 M | $1.464 \pm 0.006$ | $0.509 \pm 0.002$ | $0.500 \pm 0.005$ | $\mathbf{1.365 \pm 0.038}$ | $\mathbf{0.749 \pm 0.014}$ | $\mathbf{0.742 \pm 0.011}$ |
| HOLOPROT (◆) | 1.76 M | $1.491 \pm 0.004$ | $0.491 \pm 0.014$ | $0.482 \pm 0.017$ | $1.416 \pm 0.022$ | $0.724 \pm 0.011$ | $0.715 \pm 0.006$ |

| Model | # Params | Scaffold | | |
|---|---|---|---|---|
| | | RMSE | Pearson | Spearman |
| **Sequence-based Methods** | | | | |
| Öztürk et al. [2018] | 1.93 M | $1.908 \pm 0.145$ | $0.384 \pm 0.014$ | $0.387 \pm 0.016$ |
| Bepler and Berger [2019] | 48.8 M | $1.864 \pm 0.009$ | $0.269 \pm 0.002$ | $0.285 \pm 0.019$ |
| Rao et al. [2019] | 93.0 M | $1.680 \pm 0.055$ | $0.487 \pm 0.029$ | $0.462 \pm 0.051$ |
| Elnaggar et al. [2020] | 2.4M[1] | $1.592 \pm 0.009$ | $0.398 \pm 0.027$ | $0.409 \pm 0.029$ |
| **Surface-based Methods** | | | | |
| Gainza et al. [2020] | 0.62 M | $1.583 \pm 0.132$ | $0.416 \pm 0.111$ | $0.412 \pm 0.126$ |
| **Structure-based Methods** | | | | |
| Hermosilla et al. [2021] | 5.80 M | $1.592 \pm 0.012$ | $0.365 \pm 0.024$ | $0.373 \pm 0.019$ |
| HOLOPROT (●) | 1.44 M | $1.523 \pm 0.028$ | $0.489 \pm 0.019$ | $0.491 \pm 0.020$ |
| HOLOPROT (◆) | 1.28 M | $\mathbf{1.516 \pm 0.014}$ | $\mathbf{0.491 \pm 0.016}$ | $\mathbf{0.493 \pm 0.014}$ |

● full surface  ◆ molecular superpixels

**Evaluation metrics.** For evaluating different methods, we use three metrics – root mean squared error (RMSE), Pearson correlation coefficient, and Spearman correlation coefficient. We also include the mean and standard deviation across 3 experimental runs.

**Results.** Table 1 displays the results on protein-ligand binding affinity. HOLOPROT (●, ◆) performs consistently well across different tasks and dataset splits, outperforming all methods on the splits *scaffold* and *identity 60%*. On *identity 30%*, our method outperforms most baselines, while having lower variability across the evaluated metrics. HOLOPROT with molecular superpixels (◆) performs similar to HOLOPROT on the entire surface, with no or little performance loss, suggesting that molecular superpixels capture meaningful biological motifs. We include the models from [Townshend et al., 2020] for completeness, but note that these models were trained only using the protein binding pocket. Binding sites on proteins are often structurally highly conserved regions [Panjkovich and Daura, 2010]. Considering only binding pockets, which vary less between the train and test splits, provides an additional simplification making the task less challenging. All other baselines were tested on the full proteins.

## 5.2 Enzyme-Catalyzed Reaction Classification

Predicting the reaction class of enzymes without the use of sequence similarity allows for efficient screening of *de novo* proteins, i.e., macromolecules without evolutionary homologs, for catalytic properties [des Jardins et al., 1997]. The architecture of the classification module is described in Equation 3).

---

[1]The embeddings obtained via Elnaggar et al. [2020] were saved to disk, instead of finetuning the entire pretrained model.

[2]Equivariant neural network (ENN) on binding pocket only.

[3]Graph neural network (GNN) on binding pocket only.

Table 2: **Enzyme-Catalyzed Reaction Classification Results** Comparison of classification accuracy of HOLOPROT against other methods.

| Model | Parameters | Reaction Class Accuracy |
|---|---|---|
| **Sequence-based Methods** | | |
| Hou et al. [2018] | 41.7 M | 70.9 % |
| Bepler and Berger [2019] | 31.7 M | 66.7 % |
| Rao et al. [2019] (Transformer) | 38.4 M | 69.8 % |
| Elnaggar et al. [2020] | 420.0 M | 72.2 % |
| **Structure-based Methods** | | |
| Kipf and Welling [2017] | 1.0 M | 67.3 % |
| Derevyanko et al. [2018] | 6.0 M | 78.8 % |
| Hermosilla et al. [2021] | 9.8 M | **87.2** % |
| HOLOPROT (●) | 0.64 M | 77.8 % |
| HOLOPROT (◆) | 0.64 M | 78.9 % |

● full surface          ◆ molecular superpixels

**Dataset.** Enzyme Commission (EC) numbers constitute an ontological system with the purpose of defining and organizing enzyme functions [Webb, 1992]. The four digits of an EC number are related in a functional hierarchy, where the first level annotates the main enzymatic classes, while the next levels constitute subclasses, e.g. the EC number of the HIV-1 protease is 3.4.23.16. This task aims at predicting the enzyme-catalyzed reaction class of a protein based on according to all four levels of the EC number. We use the same dataset and splits as provided by [Hermosilla et al., 2021], comprising $37,428$ proteins from $384$ EC numbers, with $29,215$ instances for training, $2,562$ instances for validation, and $5,651$ for testing. For more details on dataset construction, we refer to Hermosilla et al. [2021, Appendix C].

**Baselines.** For the classification task, we again compare HOLOPROT against several baselines including sequence-based methods [Hou et al., 2018], methods partially pretrained on millions of sequences [Rao et al., 2019, Bepler and Berger, 2019, Elnaggar et al., 2020] as well as methods utilizing principles of geometric deep learning [Kipf and Welling, 2017, Derevyanko et al., 2018, Hermosilla et al., 2021]. The values for different baselines are taken from [Hermosilla et al., 2021].

**Evaluation metric.** Model performance is measured via the mean accuracy score.

**Results.** We report the results of enzyme-catalyzed reaction classification in Table 2. While our method (●, ◆) is unable to outperform the current state-of-the-art method [Hermosilla et al., 2021], we achieve equivalent, if not better results to other methods at a fraction of the parameters used. Molecular superpixels also capture biologically meaningful protein surface motifs, as evidenced by a small increase in the overall classification performance.

## 5.3 Ablation Studies

To further evaluate the contribution of HOLOPROT to learning multi-scale protein representations, we conduct several ablation studies. First, we analyze if the performance of the multi-scale model outperforms its isolated components, i.e. when using only structure or surface representation for subsequent downstream tasks. The second ablation axis analyzes the construction of molecular superpixel representations. Besides computing summary features for each molecular superpixel as described in Section 4, we learn patch representations via a MPN on the superpixel graph. The ablation study were conducted on both tasks, ligand binding affinity (Section 5.1) and enzyme catalytic function classification (Section 5.2).

As displayed in Table 3, HOLOPROT with (◆) and without molecular superpixels (●) improve over the performance of structure and surface representations. Further, the results of the ablation study clearly show that different protein scales are more relevant for particular downstream tasks, e.g., predicting the enzyme-catalyzed reaction class from surface only results in poor performance. We further see no

Table 3: **Ablation Studies Results** Evaluation of architectural design choices of HOLOPROT by analyzing the performance of its individual components as well as feature summarization of molecular superpixels.

| Model | Ligand Binding Affinity Sequence Identity (30 %) | | | Enzyme Class |
|---|---|---|---|---|
| | RMSE | Pearson | Spearman | Accuracy |
| Structure | $1.476 \pm 0.027$ | $0.51 \pm 0.029$ | $0.503 \pm 0.027$ | 74.2 % |
| Surface | $1.482 \pm 0.015$ | $\mathbf{0.512 \pm 0.022}$ | $\mathbf{0.505 \pm 0.017}$ | 28.6 % |
| HOLOPROT (●) | $\mathbf{1.464 \pm 0.006}$ | $0.509 \pm 0.002$ | $0.500 \pm 0.005$ | 77.8 % |
| HOLOPROT (◆) | $1.491 \pm 0.004$ | $0.491 \pm 0.014$ | $0.482 \pm 0.017$ | $\mathbf{78.9}$ % |
| HOLOPROT (■) | $1.491 \pm 0.027$ | $0.503 \pm 0.005$ | $0.492 \pm 0.004$ | 75.7 % |

●  full surface     ◆  molecular superpixels     ■  molecular superpixel with MPN

improvement in applying a MPN within a molecular superpixel (■) over using summary features (◆). Further ablation studies are presented in Appendix **??**.

## 5.4 Limitations

Despite the reported success of HOLOPROT, our method faces some limitations. First, HOLOPROT relies on existing protein structures and the corresponding generated surface manifolds. However, protein sequence data still remains the most abundant data source, and in protein design, conformations of mutated macromolecules are unknown. This limitation could however be partly remedied, (i.) by the recent advancements in protein structure prediction [Senior et al., 2020, Jumper et al., 2021, AlphaFold] [Baek et al., 2021, RoseTTAFold] and protein structure determination methods such as cryo-electron microscopy [Callaway, 2020], and (ii.) by utilizing homology modeling algorithms on available wild type structures for mutant analysis [Schymkowitz et al., 2005]. Second, our method requires precomputed surface meshes, resulting in an additional preprocessing step before deploying HOLOPROT to the desired application. This bottleneck can be bypassed by utilizing techniques developed in the concurrent work by Sverrisson et al. [2020], which allow computation and sampling of the molecular surface on-the-fly.

## 6 Conclusion

In this work, we present a novel multi-scale protein graph construction, HOLOPROT, which integrates finer and coarser representation details of a protein by connecting sequence and structure with surface. We further establish *molecular superpixels*, which capture higher-level fingerprint motifs on the protein surface, improving the memory efficiency of our construction without reducing the overall performance. We validate HOLOPROT's effectiveness and versatility through representative tasks on protein-ligand binding affinity and enzyme-catalyzed reaction class prediction. While being significantly more parameter-efficient, HOLOPROT performs consistently well across different tasks and dataset splits, partly outperforming current state-of-the-art methods. This will potentially be of great benefit and advantage when working with datasets of reduced size, e.g., comprising experiments on mutational fitness of proteins, thus opening up new possibilities within protein engineering and design, which we leave for future work.

## Acknowledgments

This project received funding from the Swiss National Science Foundation under the National Center of Competence in Research (NCCR) Catalysis under grant agreement 51NF40 180544. Moreover, we thank Mojmír Mutný and Clemens Isert for their valuable feedback.

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
