# Appendix

## A  Message Passing Network

We utilize the Weisfeiler-Lehmann network (WLN) proposed in (Lei et al., 2017) as our base message passing network. This network builds a neural equivalent of the Weisfeiler-Lehmann test for comparing graphs. For clarity, we describe the network here. Consider a graph $\mathcal{G} = (\mathcal{V}, \mathcal{E})$. Given a node $v \in \mathcal{G}$ with neighbors $N(v)$, node features $\mathbf{f}_v$ and edge features $\mathbf{f}_{uv}$ for edge $(v, u) \in \mathcal{E}$, the WLN message passing step follows as,

$$\mathbf{m}_v^{(l)} = \tau(\mathbf{U_1}\mathbf{m}_v^{(l-1)} + \mathbf{U_2} \sum_{u \in N(v)} \tau(\mathbf{V}[\mathbf{f}_u, \mathbf{f}_v])) \quad (1 \le l \le L) \tag{4}$$

where $\tau(\cdot)$ could be any non-linear function, and $L$ is the total number of message passing steps, $\mathbf{h}_v^{(0)} = \mathbf{f}_v$. The final representations for each node arise from mimicking the set comparison function in the WL isomorphism test, yielding

$$\mathbf{h}_v = \sum_{u \in N(v)} \mathbf{W}^{(0)}\mathbf{m}_u^{(L)} \odot \mathbf{W}^{(1)}\mathbf{f}_{uv} \odot \mathbf{W}^{(2)}\mathbf{m}_v^{(L)} \tag{5}$$

## B  Multi-Scale Protein Representations

### B.1  Ablation Studies

Table 4 shows ablation study results for the *identity 60%* and *scaffold* splits for PDBBIND dataset. The table will be updated with values of HOLOPROT (molecular superpixels with MPN) setting once the results for the same are available.

Table 4: **Results of the Ablation Studies** Evaluation of architectural design choices of HOLOPROT by analyzing the performance of its individual components as well as feature summarization of molecular superpixels.

| Model | Ligand Binding Affinity | | | | | |
| | Sequence Identity (60 %) | | | Scaffold | | |
| | RMSE | Pearson | Spearman | RMSE | Pearson | Spearman |
|---|---|---|---|---|---|---|
| Structure | $1.378 \pm 0.027$ | $0.738 \pm 0.014$ | $0.730 \pm 0.009$ | $1.521 \pm 0.023$ | $0.485 \pm 0.015$ | $0.492 \pm 0.013$ |
| Surface | $1.418 \pm 0.014$ | $0.719 \pm 0.005$ | $0.714 \pm 0.004$ | $1.558 \pm 0.125$ | $0.428 \pm 0.159$ | $0.429 \pm 0.181$ |
| HOLOPROT (●) | $\mathbf{1.365 \pm 0.038}$ | $\mathbf{0.749 \pm 0.014}$ | $\mathbf{0.742 \pm 0.011}$ | $1.523 \pm 0.028$ | $0.489 \pm 0.019$ | $0.491 \pm 0.021$ |
| HOLOPROT (◆) | $1.473 \pm 0.024$ | $0.667 \pm 0.011$ | $0.675 \pm 0.019$ | $\mathbf{1.517 \pm 0.014}$ | $\mathbf{0.491 \pm 0.016}$ | $\mathbf{0.493 \pm 0.014}$ |

● full surface        ◆ molecular superpixels

## C  Superpixels on Molecular Surfaces

### C.1  Computing Molecular Surfaces

Shape and surface of proteins determines their molecular interactions and thus, accurate computation of macromolecular surfaces from the provided atom point clouds is essential for elucidating their biological roles in physiological processes. A variety of methods have been proposed to compute macromolecular surfaces. *Van der Waals surfaces* is the simplest surface constructed via the topological boundary of the set of atom spheres, each of van der Waals radius of the constituent atom. However, as most of the van der Waals surface is buried in the interior of large molecules, Lee and Richards (1971) defined the solvent-accessible surface (SAS), determined by the area traced out by the center of a probe sphere as it is rolled over the van der Waals surface. Greer and Bush (1978) proposed smooth solvent-excluded surfaces (SES, or *molecular surface*) of a molecule (Connolly, 1983) defined as the boundary of the union of all possible probes having no intersection with the molecule. In this work, we utilize existing algorithm MSMS (Michel Sanner's Molecular Surface)

computing triangulated representations of the *molecular surface* relying on a reduced surface (Sanner et al., 1996).

**Details on Surface Preparation**   All proteins were triangulated using the MSMS with a hydrogen density of 3.0 and a water probe radius of 1.5. The meshes are downsampled using BLENDER (Blender Online Community, 2018) to a uniform size of roughly 2600 faces. In practice, we found that this size provided an appropriate balance between maintaining detail and memory consumption during preprocessing. Geometric and chemical features were computed directly on the protein mesh.

## C.2   Examples of Molecular Superpixels

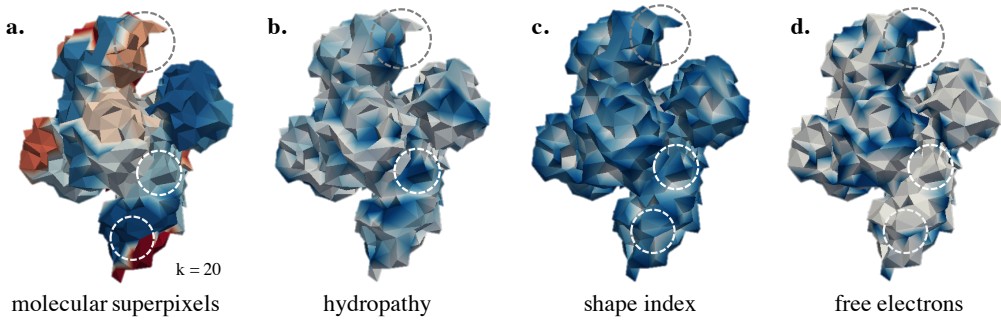

molecular superpixels     hydropathy     shape index     free electrons

Figure 3: **Molecular Superpixels and Surface Features of the Hepatitis C Virus Helicase Inhibitor (**PDB ID: 4OKS**).** Molecular superpixels, indicated by different colors ($k = 20$), and the corresponding surface features, i.e., **b.** hydropathy, **c.** shape index, and **d.** free electrons. As highlighted, molecular superpixels are spatially compact and overlap with surface regions dominated by single features such as hydrophobic patches while capturing coherent areas across all surface features. Protein complex contains 867 residues.

Additional examples of molecular superpixels and their overlap with different surface features are shown in Figure 3 and Figure 4.

## D   Experimental Details

Our model is implemented in PyTorch (Paszke et al., 2019) using the PyTorch Geometric library (Fey and Lenssen, 2019). We use the open-source software RDKit (Landrum, 2016). We used W&B (Biewald, 2020) for experiment tracking.

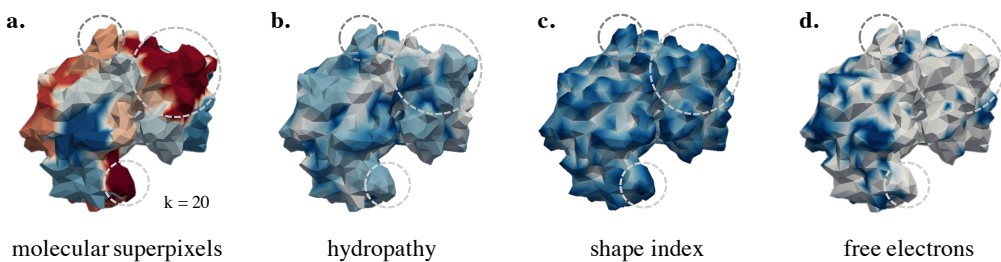

molecular superpixels     hydropathy     shape index     free electrons

Figure 4: **Molecular Superpixels and Surface Features of Endothia Aspartic Proteinase (**PDB ID: 1EPO**).** Molecular superpixels, indicated by different colors ($k = 20$), and the corresponding surface features, i.e., **b.** hydropathy, **c.** shape index, and **d.** free electrons. As highlighted, molecular superpixels are spatially compact and overlap with surface regions dominated by single features such as hydrophobic patches while capturing coherent areas across all surface features. Protein complex contains 330 residues.

## D.1 Features

### D.1.1 Surface Layer

We represent the surface layer as a graph $\mathcal{G}_\mathcal{S}$ where, for each node $u_\mathcal{S}$, we compute 4 geometric and chemical features – shape index, free electrons and proton donors, hydropathy and poisson-boltzmann electrostatics. These features are computed using code from (Gainza et al., 2020), and the binaries APBS (Baker et al., 2001), PDB2PQR (Dolinsky et al., 2007) and multivalue (provided within the APBS suite). We refer to (Gainza et al., 2020) for more details.

Two nodes share an edge if they are part of the same triangulation, and an edge is a part of two triangular faces. We compute 7 edge features – the dihedral angle between the two faces, the inner angles (one for each face) opposite to the edge, two edge-length ratios, where the edge ratio is between the length of the edge and the perpendicular (dotted) line for each adjacent face. These features were taken from (Hanocka et al., 2019). We also include the distance between the surface nodes comprising the edge and the angle between the normals at those nodes.

### D.1.2 Structure Layer

We represent the structure layer as a graph $\mathcal{G}_\mathcal{B}$ where the nodes $u_\mathcal{B}$ are the amino acid residues, and the edges occur between two amino acids within a certain distance threshold. We use the following node and edge features,

| Node Feature | Count | One-Hot | Possible Values |
|---|---|---|---|
| Residue Name | 23 | Yes | ALA, GLY etc. |
| Secondary structure the residue is part of | 8 | Yes | H, G, I, E, B, T, C, unk |
| Solvent Accessible Surface of the residue | 1 | No | - |
| Residue hydrophobicity | 1 | No | - |

As edge features, we use the angle between two residues and the distance between their $C_\alpha$ atoms. To compute the secondary structure, we use the DSSP binary (Kabsch and Sander, 1983).

### D.1.3 Ligand Molecules

We represent the ligand molecule as a graph $\mathcal{G}_\mathcal{L}$ with the following node and edge features,

| Node Feature | Count | One-Hot | Possible Values |
|---|---|---|---|
| Atom symbol | 65 | Yes | C, N, O etc. |
| Atom degree | 10 | Yes | 0, 1, 2, 3, 4, 5, 6, 7, 8, 9 |
| Implicit valence of the atom | 6 | Yes | 0, 1, 2, 3, 4, 5 |
| Explicit valence of the atom | 6 | Yes | 1, 2, 3, 4, 5, 6 |
| Part of an aromatic ring | 1 | No | 0, 1 |

| Edge Feature | Count | One-Hot | Possible Values |
|---|---|---|---|
| Bond type | 4 | Yes | Single, Double, Triple, Aromatic |
| Whether bond is conjugated | 1 | No | 0, 1 |
| Whether bond is part of ring | 1 | No | 0, 1 |

## D.2 Datasets

**Protein-Ligand Binding Affinity**   We use the refined subset of the 2019 version of PDBBind (Liu et al., 2017) and evaluate our model on 3 splits – *scaffold*, *identity 60%* and *identity 30%*. Two of these splits (*identity 60%* and *identity 30%*) are based on sequence identity, with sequences in the test set not having more than a 30% or 60% to sequences in the training set. We use the same splits as provided by (Townshend et al., 2020), and refer the reader to the same for more details on their construction. The scaffold split is prepared by computing the Bemis-Murcko scaffold (Bemis and Murcko, 1996) using RDKit for each molecule, and splitting the molecules such that molecules with rare or unseen scaffolds are part of the test set.

**Enzyme-Catalyzed Reaction Classification**   We use the PDB files and splits provided by Hermosilla et al. (2021) for this task. For more details on the dataset construction, we refer the reader to Hermosilla et al. (2021, § C).

### D.3   Network Architectures

**Hyperparameter Tuning**   For protein-ligand binding affinity prediction, we performed a hyperparameter sweep over the hidden dimensions of the surface $(150, 200, 300)$ and structure layers $(150, 200, 300)$, and the hidden dimensions of the MLP $(512, 256, [512, 256])$. For the enzyme-catalyzed reaction classification, given time constraints, our hyperparameter tuning was restricted to the learning rates $0.001, 0.0005, 0.0001$ and hidden layer activations (ReLU, LeakyReLU).

For all HOLOPROT models, we use the Adam optimizer for training and clip gradients to a maximum norm of $10.0$.

#### D.3.1   Protein-Ligand Binding Affinity

For HOLOPROT with the full surface, the surface and structure layer MPNs have hidden dimensions of $150$ and $200$ with message passing steps of $6$ and $5$ respectively. The affinity prediction MLP has a single hidden layer of dimension $512$. For HOLOPROT with molecular superpixels, the surface and structure layer MPNs have hidden dimensions of $150$ and $300$, with $5$ message passing steps. The affinity prediction MLP has a single hidden layer of dimension $256$. For both models, the ligand MPN has a hidden layer dimension of $300$, with $4$ message passing steps. We use the ReLU activation function. Starting with an initial learning rate of $0.001$, we apply a learning rate decay of $0.9$ based on a validation RMSE plateau, with an improvement threshold of $0.01$ and a patience of $5$. The HOLOPROT model for full surface has $1.44$M parameters, while the HOLOPROT model with molecular superpixels has $1.76$M parameter.

#### D.3.2   Enzyme-Catalyzed Reaction Classification

Both HOLOPROT models have a hidden dimension of $150$ for both the surface and structure layers, with $4$ and $5$ message passing steps. The classification MLP has a single hidden layer of dimension $512$. We use the LeakyReLU activation function, and apply dropout with probability $0.15$ for each message passing step, and $0.3$ for the classification MLP. Starting with an initial learning rate of $0.0005$, we apply a learning rate decay of $0.6$ based on a validation accuracy plateau, with an improvement threshold of $0.01$ and a patience of $10$. Both models have roughly $0.64$M parameters. We ran HOLOPROT with more parameters ($5.2$ M) *without* hyperparameter search due to computational restrictions, resulting in small improvement of the overall performance (i.e., an accuracy of $79.2\%$ on the enzyme-catalyzed reaction classification task).

#### D.3.3   Baselines

For protein-ligand binding affinity prediction, we use the provided code for different baselines and extend them as necessary for the task. For enzyme-catalyzed reaction classification, we use the baseline values from (Hermosilla et al., 2021).

Across all models for protein-ligand binding affinity prediction, we compute representations for proteins and ligands and concatenate them and use that as input for the MLP. For the models described in (Townshend et al., 2020), we use the reported values. Wherever possible, we restrict the ligand embedding dimension to be $300$ consistent with our experiments. The details for the remaining baselines are as follows,

**Öztürk et al. (2018)**   We downloaded the code from the official repository. The authors do not use a separate validation set, but instead use a cross-validation strategy. We combine the training and validation sets and then perform 5-fold cross-validation. The authors allow specification of hyperparameters for the number of filters $(32, 64)$, the size of ligand sequence filters $(4, 8, 12, 36)$, and size of protein sequence filters $(4, 8, 12, 36)$. We use a default batch size of $64$, and the default learning rate, and train the model for $100$ epochs. We also note that the model typically undergoes early stopping around epoch $80$. The best performing model occurred with $32$ filters, with a filter size of $4$ for ligand sequences, and $8$ for protein sequences.

**Bepler and Berger (2019)**  We downloaded the code and pretrained models from the official repository and embed the ligand sequence with a bidirectional LSTM. For embedding the protein, we use the default values as from the pretrained model. For the ligand, we use an input dimension of $512$, and a LSTM hidden dimension of $512$, with a final embedding dimension of $100$, similar to the protein. Our hyperparameter tuning is restricted to ligand input dimensions $(512, 256)$, LSTM hidden dimensions $(512, 256)$ and the MLP hidden dimensions $(512, 256, [512, 256])$. We use a batch size of $32$ and learning rate of $0.00001$ for the pretrained model, and $0.001$ for the remaining parameters.

**Rao et al. (2019)**  We downloaded the code and pretrained models from the official repository. We represent the ligand as a sequence and embed it using a Transformer, with an embedding dimension of $300$ and an intermediate size of $512$. We use a 2-layer MLP with hidden dimensions of $512$ and $256$ and dropout probabilities of $0.2$ to predict the binding affinity after concatenating the protein and ligand embeddings. For training, we use a learning rate of $0.0001$ for the pretrained model parameters and $0.001$ for the remaining parameters, and trained the model for $300$ epochs. Our hyperparameter sweep was restricted to batch size (default value 32, multiplied and divided by 2 until no improvement), and hidden layer dimensions for the MLP $(512, 256, [512, 256])$.

**Gainza et al. (2020)**  We downloaded the code from the official repository and extended the model for the ligand binding affinity task. For the protein, we use the default values provided. We represent the ligand as a graph and use the same architecture and parameters as our message passing network, with a hidden dimension of $300$. Given memory and time constraints, we were unable to perform a hyperparameter sweep.

**Hermosilla et al. (2021)**  We downloaded the code from their official repository. The proteins are embedded with an embedding dimension of $1024$. We represent the ligand as a graph and use the same architecture and parameters as our message passing network, with a hidden dimension of $300$. We concatenate the protein and ligand embeddings before using it as input for a single-layer MLP with hidden size $512$. The model is trained with the default learning rate $0.001$ and learning rate decay for $300$ epochs. Due to memory constraints, we trained with the default batch size of $8$, and performed a hyperparameter sweep for MLP hidden sizes $512, 256, [512, 256]$, and protein embedding dimensions $1024, 512$.

### D.4   Computing Infrastructure

All models were trained on a single NVIDIA 1080Ti GPU. For the PDBBind dataset, all the HOLOPROT models run within 16 hours when trained for 200 epochs, and within 8 hours when trained for 100 epochs. The baseline models (Rao et al., 2019; Hermosilla et al., 2021) take about 40 hours for running 300 epochs, while the remaining baseline models train in under 24 hours. For the enzyme-catalyzed reaction classification dataset, the HOLOPROT models are trained for 24 hours on the NVIDIA 1080Ti GPU after which it is stopped. Typically, the model is trained for 100 to 110 epochs by then, when using a batch size of 10.