# OpenReview forum: "Multi-Scale Representation Learning on Proteins"
_NeurIPS.cc/2021/Conference — NeurIPS 2021 Poster_

### Official Review · Reviewer_tk1T · 2021-07-15

**Rating:** 7
**Confidence:** 4

**Summary:**

The paper proposes a novel model incorporating sequence-level, structure-level, and surface-level information from a protein. This information is combined through an MPNN with a hierarchical structure. The authors additionally propose a molecular superpixel representation of the protein surface in order to lower computational/memory costs, with no loss of performance. The model achieves generally good results while using a very small number of parameters on both binding affinity and enzyme-catalyzed reaction classification.

**Limitations And Societal Impact:**

The authors discuss limitations of the work in the paper, and some additional suggestions are provided in my review above.

This paper is a basic computational biology work and has no immediate social impact.

**Main Review:**

This is an interesting paper in an important emerging field. Given recent advances from CASP, I expect that structure-based prediction / representation learning will play an increasingly large role in protein property prediction and protein design. This paper has two major contributions - the architecture combining structure- and surface-based features, and the use of molecular superpixels. These ideas are likely to be generally useful to the structure representation community. The evaluations are reasonable and the model achieves decent performance using relatively few parameters. The ablations also suggest both structure and surface information are useful, although the effect seems minimal in the case of ligand binding affinity.

Here are some suggestions for improvements to the paper:

- The method proposed here uses a one-hot representation of amino acid identity as the sequence level feature for each residue. Why not instead use the output of a protein language model, such as ESM-1b or ProtBERT-BFD? These feature vectors are known to contain a good deal of biologically relevant information (including e.g. secondary structure) so it seems like this is an easy way to get the benefits of a large scale language model incorporated into your model.
- In the ligand binding affinity task you do not benchmark against a recent sequence-based method (e.g. ESM-1b, ProtBERT-BFD, ProTrans-T5). It would be nice to have this evaluation.
- How was the model size for HoloProt chosen? It seem as though the number of parameters is much smaller, and you make an argument for the parameter efficiency. However, could the method be scaled up to achieve SOTA performance? In other words, was the model size chosen due to computational constraints or is there a more fundamental limitation preventing scaling the model and achieving better results?
- The paper introduces molecular superpixels as a method to reduce computational / memory costs, but the improvements are not benchmarked.
- I am hesitant to suggest additional experiments since I understand they are likely infeasible in the author response period. However, if time permits or for future publications it may be worth looking into the Atom3D suite of tasks (Townshend 2020).

Minor issue:

- The citation provided for AlphaFold2 is the Senior 2020 Nature paper describing AlphaFold1. The correct citation is found at the end of [this blog post](https://deepmind.com/blog/article/alphafold-a-solution-to-a-50-year-old-grand-challenge-in-biology).

Finally, I note that you describe training several of these models takes a significant amount of time (40 hours). For some of these models (e.g. the pretrained language models) a better option is to embed each sequence and save the features to disk (assuming you are not fine-tuning the entire model). Then during training, simply loading the features and training the MLP will likely allow you to train the model in minutes.

**Time Spent Reviewing:**

3

---

> ### Author Response · Authors · 2021-08-10
> **Language Model Baselines and Input Features**
>
> Thanks for your thorough feedback and insightful ideas!
>
> As our model depends on folded protein structures, recent breakthroughs in protein structure determination (AlphaFold 2,  RoseTTAFold) along with the released code partly remedies one limitation of our model.
> By not limiting the model to sequences and structures, but also surfaces, we extend the current perspective of protein modeling by one more axis.
> A protein's surface is particularly crucial when studying protein interface prediction to account for sterical constraints in the binding process, an application we are currently working on building upon the concepts developed in *HoloProt*.
>
> > Protein Language Model Input Features?
>
> Thank you for the suggestion, we incorporated this idea into our framework. We used the ProtBERT-BFD embeddings (Elnaggar et al. 2020) as additional node features. On both tasks, ligand binding affinity prediction and enzyme-catalyzed reaction classification, however, this did not result in improved performance, with the performance being very similar to Elnaggar et al. (2020). This suggests that the sequence based embeddings were dominating the training process.
> We will further analyze this and investigate the effect of using sequence embeddings as additional node features.
>
> > Additional Benchmarks.
>
> We ran ProtBERT-BFD (Elnaggar et al., 2020) on the binding affinity task. While always performing worse than *HoloProt*, it indeed performs better than other language model baselines (see Table below).
>
> | Model          |                   | Identity 30% |                   |                   |Identity 60% |                   |
> |------------------------|-------------------|-------------------------|-------------------|-------------------|-------------------------|-------------------|
> |                        | RMSE              | Pearson                 | Spearman          | RMSE              | Pearson                 | Spearman          |
> | Elnagger et al. (2020) | 1.544 $\pm$ 0.015 | 0.438 $\pm$ 0.053       | 0.434 $\pm$ 0.058 | 1.641 $\pm$ 0.016 | 0.595 $\pm$ 0.014       | 0.588 $\pm$ 0.009 |
> |          |                   | **Scaffold**       |                   |
> |                        | RMSE              | Pearson                 | Spearman          |
> | Elnagger et al. (2020) | 1.592 $\pm$ 0.009 | 0.398 $\pm$ 0.027       | 0.409 $\pm$ 0.029 |
>
> > Model Size.
>
> The main constraint of the project was limited computational resources, which determined our model size. To test the effect of model size on the accuracy, we train the current state-of-the-art method by Hermosilla et al. (2021) with a similar number of parameters as our model (reduction of 10x compared to the original paper). The model performance is around 78%, showing a reduction in classification accuracy compared to the large-scale version. Based on this experiment, we expect our model to also show an improvement in performance with more parameters, which we hope to be able to test in the near future.
>
> > Memory Improvement of Molecular Superpixels.
>
> Molecular superpixels provide on average a 3x memory footprint reduction compared to the surface graph. To avoid CUDA memory issues even with low batch sizes, *HoloProt* without superpixels requires standard compression techniques not designed originally for molecular surfaces. With molecular superpixels, however, we can segment arbitrarily detailed surfaces without relying on these compression techniques, and without loss of performance as indicated by our results.
> The increased batch size further accelerates training. We will add these additional explanations to the Appendix.
>
>
> > Wrong Citations.
>
> Thanks for pointing out the flaw in the reference, we will of course correct this.

---

> > ### Comment · Reviewer_tk1T · 2021-08-22
> > **Response to authors**
> >
> > Thank you for the additional benchmarks and clarifications. Overall, I like the ideas presented here and I think they are generally useful to the community. The only thing preventing me from enthusiastically recommending acceptance is the current results.
> >
> > It looks like all results presented were trained on a single 1080Ti GPU. Given the limited computational resources, it is understandable that the authors have some trouble scaling the model. The comparison to Hermosilla et al. with reduced parameters is interesting, but not quite the same as scaling the model proposed here to match. If this model had 40M or 100M parameters, I would not ask the authors to scale. But 10M parameters is not that large. There are several strategies (e.g. gradient checkpointing, gradient accumulation) which could be used to mitigate GPU memory issues. It seems like a scaled version of this model could be trained in ~1 week on the authors’ compute setup based on the timings for the current version reported. If I am mistaken in this estimate, please let me know how long you estimate it would take. It may be that this experiment cannot be carried out before the author response deadline, but could be carried out before the camera ready deadline.
> >
> > Overall, I still lean towards accepting this paper even if results are not SOTA on current benchmarks. The ideas are compelling and the contribution is novel. However, scaling the method and/or showing clear improvement on some other task (e.g. the directed evolution application mentioned in the author response) would make it easier to advocate for the method.

---

> > > ### Author Response · Authors · 2021-08-25
> > > **Scaling up HoloProt**
> > >
> > > Thank you for your valuable comments! We completely agree that scaling down would not be the same as scaling up, and just wanted to use the smaller experiment as a means to understand model scalability. The current bottlenecks on CUDA are largely caused by the gradient caches across the surface and structure layers. When scaling up to 10M parameters, we expect the following changes: i) Reduced batch size (1 or 2 instead of 10) with gradient accumulation similar to our setup for PDBBind ii) Training the model for longer (~300 epochs as Hermosilla et al.), and iii) Hyperparameter tuning. Running an initial experiment on a 5M parameter model with batch size 1 (and gradient accumulation every 32 steps), it takes the model about 70 mins to complete one epoch, which would roughly translate to 14.5 days to complete 300 epochs, or a week for 150 epochs (similar to the reviewer's estimate), not taking into account additional time to conduct hyperparameter tuning on the larger dataset. As running this set of experiments would have exceeded the response period, we instead concentrated on adding additional baselines and running experiments exploring the scalability of the model.
> > >
> > > We will, as suggested by the reviewer, include results for larger number of parameters in the camera ready version.

---

> > > > ### Comment · Reviewer_tk1T · 2021-09-01
> > > > **update score to accept**
> > > >
> > > > Given author responses, on balance I think it is better to accept than reject this paper, so I am raising my score to 7.

---

### Official Review · Reviewer_hzxn · 2021-07-15

**Rating:** 7
**Confidence:** 4

**Summary:**

The authors introduce a representation learning approach to protein structures which encompasses primary sequence, secondary structure, inter-residue distances, and surface features captured by triangulations. These different scales are encoded using two linked graph representations, and message passing networks are trained to learn geometric features from this system. Applications to protein-ligand binding and enzyme reaction classification provide evidence the model can learn effectively with few parameters.

**Limitations And Societal Impact:**

1. Choice of language model in Table 1 could be improved. Either use of ProtBERT-BFD (Elnaggar et al 2020) or ESM (Rives et al 2020) are appropriate. Given the use of Elnaggar et al in Table 2, this might be the simpler choice.
2. The change in performance from structure-only to Holoprot across Table 3 is favorable to Holoprot but not drastic. This seems to suggest that the addition of surface-level representation is improving performance in specific regimes or subproblems. Understanding this phenomenon would be very useful for future multiscale representation learning. Currently the authors claim "HOLOPROT with and without molecular superpixels improve over the performance of structure and surface representations" but this claim seems a bit strong, especially given the numbers on protein-ligand binding.
3. The authors do not provide any suggestion for why they lose to Townshend et al at 30% sequence identity splits but beat them otherwise. Is this due to over-reliance on sequence features by Holoprot? Again, some analysis would be illuminating here.


**Main Review:**

This paper pursues an interesting and novel direction in protein representation learning, seeking to integrate multiple existing approaches to feature extraction in a single architecture. Given the plethora of sequence and structure-based models, as well as recent work on surface representation, this is a natural line of investigation to pursue. The execution is simple and clean, making the paper easy to follow. Evaluations and empirical results are good but improvements or further analysis will strengthen the overall work.

Strengths:
1. The features chosen at each input scale are well-chosen. The graph structures representing each scale and the model choices for linking these scales are also reasonable.
2. Applying superpixels to protein surfaces is an interesting idea in its own right, and the authors systematically evaluate its impact on performance throughout the paper. The particular choice of superpixel generation algorithm is motivated well by the authors.
3. Dataset choices were good and proper attention was paid to splitting of train/eval/test sets.
4. The ablation study in Table 3 is thorough and provides context for model performance.

Weaknesses (elaborated on in Limitations):
i. The language model baseline for protein-ligand binding affinity needs to be improved. Given that ProtBERT-BFD (Elnaggar et al 2020) is included in Table 2, this would be a natural choice to include for Table 1 as well. This model has proven substantially better than previous language models (such as Rao et al or Bepler and Berger), so it is necessary to get an informative comparison to pretrained sequence models.
ii. A number of analysis questions should be explored further, as they are central to the story. Major ones include: why is structure-only in Table 3 comparable to Holoprot across all tasks? why does Holoprot do worse on 30% sequence identity splits but better on 60% and scaffold-based splits in Table 1?
iii. The authors argue their model is more parameter efficient in settings where it loses to the Hermosilla et al baseline. Is it instead possible for them to increase the parameters of Holoprot by an order of magnitude and see if this improves performance?

Minor comments:
- The formatting of Table 2 could be improved to highlight which methods are sequence-only, which are structure-only, etc.

Things that would raise my score:
1) Inclusion of ProtBERT-BFD for the protein-ligand binding results.
2) Expanded analyses identifying successes or challenges of the model. Some are suggested in this review, but these are just suggestions. This is particularly valuable for future work from the authors or others in the community.

**Time Spent Reviewing:**

3

---

> ### Author Response · Authors · 2021-08-10
> **Added Language Model Baseline and Further Explanations**
>
> Thanks for your thorough and valuable feedback! We have added additional baselines and expanded the analysis of *HoloProt* as suggested by the reviewer:
>
> > Additional Benchmarks.
>
> We ran ProtBERT-BFD (Elnaggar et al. 2020) on the binding affinity task. While always performing worse than *HoloProt*, it indeed performs better than other language model baselines (see Table below).
>
> | Model          |                   | Identity 30% |                   |                   |Identity 60% |                   |
> |------------------------|-------------------|-------------------------|-------------------|-------------------|-------------------------|-------------------|
> |                        | RMSE              | Pearson                 | Spearman          | RMSE              | Pearson                 | Spearman          |
> | Elnagger et al. (2020) | 1.544 $\pm$ 0.015 | 0.438 $\pm$ 0.053       | 0.434 $\pm$ 0.058 | 1.641 $\pm$ 0.016 | 0.595 $\pm$ 0.014       | 0.588 $\pm$ 0.009 |
> |          |                   | **Scaffold**       |                   |
> |                        | RMSE              | Pearson                 | Spearman          |
> | Elnagger et al. (2020) | 1.592 $\pm$ 0.009 | 0.398 $\pm$ 0.027       | 0.409 $\pm$ 0.029 |
>
>
> > Extended Analysis.
>
> **Low Performance on Sequence-Identity 30%**
>
> For the 30% sequence identity split, proteins in the test set do not bear more than a 30 % sequence similarity to proteins in the training set. Besides the split based on the ligand characteristics (scaffold), this is one of the harder evaluation settings.
> In order to investigate the performance differences compared to Townshend et al. (2020), we had a closer look at their codebase.
> The performance reported in the paper corresponds to binding affinity predicted using only the binding pocket
> (see [ENN](https://github.com/drorlab/atom3d/blob/d87369e315eb68a1b17bb95dfd3447f4421a15a3/examples/lba/enn/data.py#215) and [GNN](https://github.com/drorlab/atom3d/blob/57014866855bf8935fba0eba2de332e6c45d29c9/examples/lba/gnn/data.py#L18)) instead of the full protein, as we assumed previously. Binding sites on proteins are often structurally highly conserved regions (Panjkovich et al. BMC Structural Biology, 2010). Considering only binding pockets, which vary less between the train and test splits, provides an additional simplification making the task less challenging. All other baselines were tested on the full proteins.  In practical applications like de novo protein design screened for strong affinity to potential downstream targets, the binding pockets are generally unknown and thus evaluating and improving performance using the full protein presents a more realistic challenge.  We are currently evaluating Townshend et al. (2020)'s ENN and GNN on the full protein. The training, however, is very slow and we do encounter CUDA out of memory issues. We refer to their paper for details on size restrictions for the ENN.
>
> Further, *HoloProt* does not exhibit an overreliance on sequence features. Beyond one-hot encodings of the amino acids, we do not use additional sequence features such as those based on multiple-sequence alignments, e.g., BLOSUM.
>
> **Influence of Model Size.**
>
> The main constraint of the project was limited computational resources, which determined our model size. To test the effect of model size on accuracy, we train the current state-of-the-art method by Hermosilla et al. (2021) with a similar number of parameters as our model (reduction of 10x compared to the original paper). The model performance is around 78%, showing a reduction in classification accuracy compared to the large-scale version. Based on this experiment, we expect our model to also show an improvement in performance with more parameters, which we hope to be able to test in the near future.
>
>
> **Influence of Structure and Surface to Overall Model-Performance.**
>
> For the binding affinity task, both structure and surface serve as useful representations, as indicated by their performance within the confidence intervals of each other (Table 3 and 4 in Appendix). This makes sense as the shape of the binding pocket in addition to charges or other stereochemical properties plays an important role in the binding strength. However, for enzyme classification, we noticed that two enzyme members of the same reaction class can have differing surfaces despite only minor differences in the sequence. Therefore, methods utilizing protein sequence (e.g.,  based on multiple sequence alignments) or structure are good predictors of the corresponding enzyme class, but the surface in itself is not, which also agrees with the experimental and ablation results. From Table 3 and 4 (in the Appendix), a holistic representation combining multiple scales offers performance improvements over individual scales and demonstrates a more consistent training behavior across different metrics (e.g., Pearson and Spearman correlation as well as RMSD).
>
>
> > Presentation of Results.
>
> We have improved the presentation of the results by grouping the methods in Table 1 and 2 based on input data types and by adding the model type mentioned in the text additionally into the tables.

---

> > ### Comment · Reviewer_hzxn · 2021-08-29
> > **Increasing score**
> >
> > Thanks to the authors for such thorough responses to all of my previous points. I am happy to raise my score to a 7 based on the additional material and discussion above.

---

### Official Review · Reviewer_TKtD · 2021-07-19

**Rating:** 6
**Confidence:** 4

**Summary:**

The authors consider the problem of learning protein representations which jointly leverage features at multiple scales: sequence, structure, and surface. To accomplish this, a graph construction is proposed with interconnected structure and surface layers, and physicochemical node/edge features. A proposed model encodes the graph sublayers into a unified representation of the protein.  Computed representations are evaluated on two tasks: prediction of (1) ligand binding affinity and (2) enzyme class.  Finally, the authors propose the notion of “molecular superpixels” - adapting an image segmentation algorithm to partition molecular surfaces into compact, geometrically and chemically homogenous zones.

**Limitations And Societal Impact:**

Yes.  We appreciate the acknowledgement in [section 5.4 - Limitations] and [section 2 - Related Work] that baseline models were trained with very divergent input data modalities, and that this method explicitly requires structures.

**Main Review:**


The paper presents a new approach to multi-scale protein representation learning, with a creative adaptation of superpixels for partitioning molecular surfaces into compact, homogenous regions.

Main Strengths:



* The multi-scale graph construction - uniting sequence, structure, and surface - is well motivated, conceptually simple, and a novel contribution.
* The application of superpixels to biomolecules is a novel contribution, and is compelling for: (1) its connection to a successful method in an outside field, and (2) its general applicability to any feature set.

Major Weaknesses:



* Final result impact is diluted by underperformance relative to (Hermosilla et al, 2021) on enzyme class task, despite Holoprot’s additional inclusion of surface features.
* Results in Tables 1 and 2 are hard to parse without indicators of model type and input data (sequences, structures), which are important to contextualize results.  The results are less impressive when ignoring the sequence-only rows and including other strong baselines in the Hermosilla baseline source: HHSuite (82.6%) and Rao - LSTM (79.9%).

Based on the novelty + conceptual simplicity of the proposed method, but with the weakness of experimental results limiting the impact: I recommend 6: Marginally above acceptance threshold.


### Setup, approach, originality



* Problem setup is good (forward-looking and impactful): structure-based representation + surface graph, evaluated on reasonable tasks.
    * [-]  But, rather than tasks relying on fixed-size embeddings of the full protein, predicting protein surface interaction location like the tasks in Gainza et al 2020 seems more promising and appropriate for this model class - was this considered?
* The multi-scale graph construction is conceptually simple, and well motivated.
* The application of superpixels to biomolecules is a novel contribution, and is compelling for its simplicity and generality.
* Though, as the author’s allude to in section [4 - Superpixels on Molecular Surfaces] there is a long history of methods for identifying regions on three-dimensional molecular surfaces.  Molecular superpixels are stated to be advantageous due to better geometric compactness relative to some baselines, but no quantitative analysis, nor claim is made of general superiority over alternative algorithms - fair.


### Quality

Major:



* [+] Ablations were informative.  It is nice to see that more powerful pooling on molecular superpixels was unnecessary.  And it is interesting that the marginal benefit of the structure / surface layers is so different across tasks.
* [+] [5.4 - Limitations]  Important call-out that this method requires structures is appreciated.
* [?] [Abstract]  It is claimed that Holoprot is more _consistent_ than other methods. How is this reflected in the results? The level of variation looks comparable to that of Hermosilla/Gainza/others.
* [-] The list of features in the surface and structure layers node/edge features seem extensive.
    * What is the reasoning behind the node/edge features chosen in the structure layer?
    * It would be informative to know how helpful the 7 edge features in the surface network (Hanocka et al 2019) are.  Since it seems Gainza did not use these. (e.g. Are features or the model the driver of the boost over Gainza?)

Minor:



* [Table 1] Townshend et al. (2020) results are missing for the scaffold-level split?
* Line 540: “structure” is a typo.

### Clarity


Major:


* [+] Paper + supplement are well organized.  Explanations are clear, and the appendix concisely adds much useful detail.
* [Table 1 + 2]
    * The tables would benefit highly from organization by model / input-data type, as opposed to the current organization by chronology.
        * Partial example: Table 1 of the baseline results source (Hermosilla, et al., 2021), which specifies architecture + special notes on data sources.
    * [-] Other top baselines from that paper should not be omitted: HHSuite (82.6%) and Rao - LSTM (79.9%).
* The superpixel ERS method is hard to follow, could probably be expanded in appendix, and a high level intuitive explanation be provided in the maintext.

### Significance

* Multi-scale graph construction and representation algorithms are conceptually clean and well motivated.  The focus on explicitly incorporating protein surface features seems a useful and forward-looking approach.
* The superpixel algorithm is not quantitatively shown as superior to alternative algorithms, but is compelling for its simplicity and general applicability.  Future researchers may find this to be a useful primitive.
* The final result impact is diluted by underperformance relative to (Hermosilla et al, 2021) on the enzyme class task, despite Holoprot’s additional inclusion of surface features.
* Low data regime argument is interesting [Conclusion], but could be improved:
    * Is it possible to scale the current model, for better performance?
    * Is it possible to evaluate on a task that is more in the low data regime?

**Time Spent Reviewing:**

9

---

> ### Author Response · Authors · 2021-08-10
> **New Baselines, Improved Presentation of Results and Future Applications**
>
> Thanks for your thorough feedback and insightful ideas!
>
> The suggestion of the reviewer for future work is indeed quite spot on. We are applying *HoloProt* in the low data regime for a directed evolution application. Here the aim is to infer the effects of mutations introduced via combinatorial libraries, potentially automating large-scale screening. While single mutations often have only minor effects on a protein's structure, the corresponding change on a protein's surface is significant.
> While demonstrating the applicability of *HoloProt* to standard tasks as presented in this paper, integrating all scales into the model will also be beneficial for applications such as mutagenesis analysis (often in the low data regime where parameter efficiency is crucial). We are currently working on this application as future work.
>
>
> > Chosen Input Features.
>
> For the structure layer, our node and edge features were largely informed by Fout et al. (2017). Compared to Gainza et al. (2020), we think the model provides the performance boost, as can be seen from the performance of HoloProt with molecular superpixels which are better or within the confidence intervals of Gainza et al. (2020). The superpixel graphs do not use any edge features from Hanocka et al. (2018), both in the segmentation and the graph construction.
>
> > Consistent Performance of *HoloProt*.
>
> On the regression task and across different splits, *HoloProt* finds better solutions while maintaining a similar variation to other baselines. We will add this clarification in the updated version.
>
> > Model Size.
>
> The main constraint of the project was limited computational resources, which determined our model size. To test the effect of model size on accuracy, we train the current state-of-the-art method by Hermosilla et al. (2021) with a similar number of parameters as our model (reduction of 10x compared to the original paper). The model performance is around 78%, showing a reduction in classification accuracy compared to the large-scale version. Based on this experiment, we expect our model to also show an improvement in performance with more parameters, which we hope to be able to test in the near future.
>
> > Additional Baselines.
>
> We added additional baselines to the evaluation section requested by the reviewer, including the LSTM by Rao et al.(2019) and HHSuite by Hou et al. (2018).
> In order to investigate the performance differences compared to Townshend et al. (2020), we had a closer look at their codebase.
> The performance reported in the paper corresponds to binding affinity predicted using only the binding pocket
> (see [ENN](https://github.com/drorlab/atom3d/blob/d87369e315eb68a1b17bb95dfd3447f4421a15a3/examples/lba/enn/data.py#215) and [GNN](https://github.com/drorlab/atom3d/blob/57014866855bf8935fba0eba2de332e6c45d29c9/examples/lba/gnn/data.py#L18)) instead of the full protein, as we assumed previously. Binding sites on proteins are often structurally highly conserved regions (Panjkovich et al. BMC Structural Biology, 2010). Considering only binding pockets, which vary less between the train and test splits, provides an additional simplification making the task less challenging. All other baselines were tested on the full proteins.  In practical applications like de novo protein design screened for strong affinity to potential downstream targets, the binding pockets are generally unknown and thus evaluating and improving performance using the full protein presents a more realistic challenge.  We are currently evaluating Townshend et al. (2020)'s ENN and GNN on the full protein. The training, however, is very slow and we do encounter CUDA out of memory issues. We refer to their paper for details on size restrictions for the ENN.
>
> > Alternative Algorithms for Superpixels.
>
> The choice of ERS was made based on the comparison by Stutz et al. (2018), where ERS was the best performing method applicable to general surfaces across different tasks, as well as in initial experiments by us on molecular surfaces.
> We additionally explored end-to-end differentiable solutions such as Hanocka et al. (2018) and Bednarik et al. CVPR (2020), which significantly increased the computational cost and speed. None of the tested methods even achieved the computation of compact and homogeneous clusters, and both methods were very tricky to tune.
>
> > Presentation of Results.
>
> We have improved the presentation of the results by grouping the methods in Table 1 based on input data types and adding the model type mentioned in the text additionally into the tables. We will furthermore extend the introduction and description of molecular superpixels in the Appendix.

---

> > ### Comment · Reviewer_TKtD · 2021-08-26
> > **Response to authors**
> >
> > Thank you for the comments and  additional clarifications. Specifically
> > > HOLOPROT with molecular superpixels (..) superpixel graphs do not use any edge features from Hanocka et al
> >
> > had not been clear to me from the Section 5.1 discussion.
> > With some more careful cross-checking of table 1, I narrowly tend to agree that
> >
> > > Compared to Gainza et al. (2020), we think the model provides the performance boost
> >
> > given that the harder splits (seqid60% and scaffold splits) there seems to be a narrow gain in Table 1 to HOLOPROT molecular superpixels, although it is mostly within the noise levels.
> >
> > All in all, I stay with the original assessment which is strongly in line with the other reviewers' view on this paper.

---

### Decision · Program_Chairs · 2021-09-27

**Decision:**

Accept (Poster)

**Comment:**

The paper proposes a very interesting approach for protein representation learning. The AC and reviewers greatly appreciated the author feedback and we urge the authors to incorporate their points into the manuscript.

In particular it would be important to include the comparison with ProtBERT-BFD  and the extended analysis results.

If the authors cannot include results for larger number of parameters as promised, it is critical that they include comments on limited resources, where the bottlenecks are and path to resolve these.

---

> ### Public Comment · ~Fang_Wu1 · 2022-01-23
> **Confusion about the published version**
>
> Hi, this is a great paper that provides a brand new perspective to deal with proteins. However, as pointed out by most reviewers and AC, it is necessary to include the comparison with ProtBERT-BFD and other analysis results. But unfortunately, I did not see any updated information regarding this part.

---

> > ### Public Comment · ~Brian_Wiley1 · 2024-03-27
> > **Confused about comment**
> >
> > Hi,
> >
> > I eventually will be submitting papers on using protein conformation from molecular dynamics systems and biophysical laws of molecular interactions.  I am wondering why this paper needs to compare itself to ProtBERT-BFD which complete disregards any biophysical interactions of proteins and the sum of their parts.  Their embeddings are simply at a per residue level but individual residues are not in a vacuum by themselves.  This would be like taking each part of a car in a longitudinal dynamics study to train for self-driving but using on the properties of each part individually, removing all road conditions (say the protein solvent), the connections between the parts (force field parameters of bonded and non-bonded atoms), how different people drive the car, shape of the car, width of road, speed/force, what the car is carrying:
> >
> > shape/size/aerodynamics of car -> oligomerization/shape/pockets/cavities/curvature or flatteness/charge of protein
> >
> > width of road -> contraints of protein
> >
> > speed/force of car -> velocity and force of atoms/residues in protein
> >
> > what care is carrying -> post-translations modification of protein
> >
> > Comparing biophysical protein models to protein language models is like comparing duck-confit to spagetti!